# *ABL* Genomic Editing Sufficiently Abolishes Oncogenesis of Human Chronic Myeloid Leukemia Cells In Vitro and In Vivo

**DOI:** 10.3390/cancers12061399

**Published:** 2020-05-29

**Authors:** Shu-Huey Chen, Yao-Yu Hsieh, Huey-En Tzeng, Chun-Yu Lin, Kai-Wen Hsu, Yun-Shan Chiang, Su-Mei Lin, Ming-Jang Su, Wen-Shyang Hsieh, Chia-Hwa Lee

**Affiliations:** 1Department of Pediatrics, School of Medicine, College of Medicine, Taipei Medical University, Taipei 11031, Taiwan; Shu117@tmu.edu.tw; 2Department of Pediatrics, Shuang Ho Hospital, Taipei Medical University, New Taipei City 23561, Taiwan; 3Division of Hematology and Oncology, Shuang Ho Hospital, Taipei Medical University, New Taipei City 23561, Taiwan; 10573@s.tmu.edu.tw; 4Division of Hematology and Oncology, Department of Internal Medicine, School of Medicine, College of Medicine, Taipei Medical University, Taipei 11031, Taiwan; 5Ph.D. Program for Cancer Molecular Biology and Drug Discovery, College of Medical Science and Technology, Taipei Medical University, Taipei 11031, Taiwan; 161016@h.tmu.edu.tw; 6Graduate Institute of Cancer Biology and Drug Discovery, College of Medical Science and Technology, Taipei Medical University, Taipei 11031, Taiwan; 7Division of Hematology/Oncology, Department of Medicine, Taipei Medical University Hospital, Taipei 11031, Taiwan; 8Institute of Bioinformatics and Systems Biology, National Chiao Tung University, Hsinchu 30068, Taiwan; chunyulin@nctu.edu.tw; 9Center for Intelligent Drug Systems and Smart Bio-devices, National Chiao Tung University, Hsinchu 30068, Taiwan; 10Institute of New Drug Development, China Medical University, Taichung City 40402, Taiwan; kwhsu@mail.cmu.edu.tw; 11Research Center for Cancer Biology, China Medical University, Taichung City 40402, Taiwan; 12School of Medical Laboratory Science and Biotechnology, College of Medical Science and Technology, Taipei Medical University, Taipei 11031, Taiwan; yuc114@pitt.edu; 13Department of Pharmaceutical Sciences, School of Pharmacy, University of Pittsburgh, Pittsburgh, PA 15213, USA; 14Department of Pathology and Laboratory Medicine, Shin Kong Wu Ho-Su Memorial Hospital, Taipei 11101, Taiwan; sumeinancy@gmail.com; 15Department of Clinical Pathology, Shuang Ho Hospital, Taipei Medical University, New Taipei City 23561, Taiwan; 19021@s.tmu.edu.tw; 16Department of Family Medicine, Shuang Ho Hospital, Taipei Medical University, New Taipei City 23561, Taiwan; 17Department of Laboratory Medicine, Shuang Ho Hospital, Taipei Medical University, New Taipei City 23561, Taiwan; wshiehms2@gmail.com; 18Ph.D. Program in Medical Biotechnology, College of Medical Science and Technology, Taipei Medical University, Taipei 11031, Taiwan; 19TMU Research Center of Cancer Translational Medicine, Taipei 11031, Taiwan

**Keywords:** CRISPR/Cas9, gene edit, Philadelphia chromosome, BCR-ABL, CML

## Abstract

Chronic myelogenous leukemia (CML) is the most common type of leukemia in adults, and more than 90% of CML patients harbor the abnormal Philadelphia chromosome (Ph) that encodes the BCR-ABL oncoprotein. Although the ABL kinase inhibitor (imatinib) has proven to be very effective in achieving high remission rates and improving prognosis, up to 33% of CML patients still cannot achieve an optimal response. Here, we used CRISPR/Cas9 to specifically target the *BCR*-*ABL* junction region in K562 cells, resulting in the inhibition of cancer cell growth and oncogenesis. Due to the variety of *BCR*-*ABL* junctions in CML patients, we utilized gene editing of the human *ABL* gene for clinical applications. Using the *ABL* gene-edited virus in K562 cells, we detected 41.2% indels in *ABL* sgRNA_2-infected cells. The *ABL-*edited cells reveled significant suppression of BCR-ABL protein expression and downstream signals, inhibiting cell growth and increasing cell apoptosis. Next, we introduced the *ABL* gene-edited virus into a systemic K562 leukemia xenograft mouse model, and bioluminescence imaging of the mice showed a significant reduction in the leukemia cell population in *ABL*-targeted mice, compared to the scramble sgRNA virus-injected mice. In CML cells from clinical samples, infection with the *ABL* gene-edited virus resulted in more than 30.9% indels and significant cancer cell death. Notably, no off-target effects or bone marrow cell suppression was found using the *ABL* gene-edited virus, ensuring both user safety and treatment efficacy. This study demonstrated the critical role of the *ABL* gene in maintaining CML cell survival and tumorigenicity in vitro and in vivo. *ABL* gene editing-based therapy might provide a potential strategy for imatinib-insensitive or resistant CML patients.

## 1. Introduction

Leukemia is classified as acute or chronic and as myelogenous or lymphocytic; the subtypes include acute myelogenous leukemia (AML), chronic myelogenous leukemia (CML), acute lymphocytic leukemia (ALL) and chronic lymphocytic leukemia (CLL), and chronic leukemia grows slowly, and progressively worsens over time. CML is the most common type of leukemia in adults, comprising 15%–25% of all adult leukemia cases worldwide [1]. A chromosome translocation, between the long arms of chromosomes 9 and 22, t(9;22)(q34;q11), is found in over 90% of CML patients, in a lower proportion of ALL or biphenotypic acute leukemia cases and in rare cases of de novo AML [2,3]. This well-known Philadelphia (Ph) chromosome produces the BCR-ABL oncogenic fusion protein that activates multiple signaling pathways involved in the cell cycle, adhesion and apoptosis [4,5]. In addition, expression of this BCR-ABL oncoprotein transforms hematopoietic progenitor cells; in an animal model, this transformation event activated downstream signaling proteins that increase cell survival and proliferation, indicating the essential oncogenic role of BCR-ABL in CML cells [6].

There are several treatments for CML patients, including tyrosine kinase inhibitors (TKIs), hematopoietic stem cell transplantation and chemotherapy. The predominant treatment for CML is a TKI, and the TKI imatinib (Gleevec) is the first-line targeted therapy for CML. Imatinib has been shown in recent years to be highly effective at increasing the life expectancy of CML patients [7]. Imatinib inhibits BCR-ABL, subsequently inhibiting the proliferation of abnormal leukemia cells [8]. However, this TKI has side effects, such as nausea, headache, diarrhea, fatigue, rash, hypertension and diabetes. Additionally, strict treatment compliance is essential, since any missed dose might contribute to the development of drug resistance and gene mutations [9], leading to leukemia recurrence. Accordingly [10], approximately 33% of patients with CML treated with imatinib do not achieve a complete cytogenetic response (CCyR). Therefore, resistance to TKIs is still the primary problem that needs to be solved in CML treatment.

The mechanisms of imatinib resistance could be both BCR-ABL dependent (gene amplification or point mutations) and BCR-ABL independent [11]. Therefore, a more effective and precise CML therapeutic strategy is urgently needed. Recently, clustered regularly interspaced short palindromic repeats (CRISPR)/Cas9 technology has initiated a new era of genome editing. CRISPR/Cas9 technology overcomes the limitations of traditional genome editing techniques, which are considered ineffective, time-consuming and laborious; thus, it can be applied as a general-purpose gene editing system [12]. CRISPR/Cas9 generates double-strand breaks (DSBs) at target sites by recognizing 20-nt sequences that match an engineered gRNA and a 3-nt protospacer adjacent motif (PAM), located downstream of the target sequence. The subsequent cellular DNA repair process, nonhomologous end joining (NHEJ), is an error-prone DSB repair mechanism that introduces the desired genetic insertions, deletions or substitutions at the target site [13]. In other words, CRISPR/Cas9 allows researchers to rapidly generate a pool of gene knockout cells without a massive gene engineering design effort. This gene editing system has been widely used to correct mutated genes related to disease and cancer [14]. The CRISPR/Cas9 system can potentially modify disease-related genes in vitro and in vivo; however, few studies have explored the potential of this system in general cancer therapeutic applications.

The *BCR*-*ABL* fusion gene is an ideal target for CRISPR/Cas9 gene therapy in CML [15]. However, the junction regions of the *BCR*-*ABL* gene are different in every CML patient [16]. Therefore, we utilized the CRISPR/Cas9 gene editing strategy to cleave the *ABL* gene and eliminated its oncogenic activity in vitro. To ensure gene editing efficiency, we used several assays, such as Sanger DNA sequencing, tracking of indels by decomposition (TIDE) analysis, restriction fragment length polymorphism (RFLP) of the *ABL* gene region and protein analysis of K562 cells. In addition, the safety of CRISPR/Cas9-mediated gene editing in human cells was addressed by an analysis of potential off-target genes and bone marrow cells. Notably, our effective anticancer results in a systemic leukemia animal model treated with virus-mediated gene editing therapy suggested an alternative treatment for clinical CML patients who are insensitive or resistant to imatinib treatment.

## 2. Materials and Methods

### 2.1. Cell Culture and Patient Samples

The human leukemia K562 cell line (CML) was kindly provided by Dr. Kai-Wen Hsu, Research Center for Tumor Medical Science, China Medical University, Taichung, Taiwan. The bone marrow derived epithelial cells were kindly proved by Dr. Chia-Ling Hsieh, The Ph.D. Program for Translational Medicine, College of Medical Science and Technology, Taipei Medical University, Taipei, Taiwan. The cells were maintained in Dulbecco’s Modified Eagle Medium: Nutrient Mixture F-12 (DMEM/F-12) (Gibco, Grand Island, NY, USA). The peripheral blood of CML participants and healthy controls was obtained at Shuang Ho Hospital, Taipei Medical University, New Taipei City, Taiwan, according to a protocol approved by the Institutional Review Board (N201711069). Clinical parameters, such as RBC count, WBC count, karyotype and fluorescent in-situ hybridization (FISH) analysis, were determined.

### 2.2. MTT Cell Viability and BrdU Cell Proliferation Assay

Cell viability was determined using the 3-(4,5-dimethylthiazol-2-yl)-2,5-diphenyltetrazolium (MTT), which is based on reduction of the yellow MTT to purple formazan by living cells [17]. In 96-well plates, 8 × 10^4^ cells were seeded in 100 μL of DMEM/F12 per well and were exposed to different concentrations of Imatinib according to the experimental protocol. After 48 h of treatment, the medium was changed to fresh medium containing 1 mg/mL of MTT. Two hours later, 100 μL of DMSO was added in each well and the absorbance at 570 and 630 nM was determined. The percentage of cell viability was calculated using a formula [percentage viability = (average OD of sample/average OD of control) × 100].

K562 cell proliferation was determined using the colorimetric bromodeoxyuridine (BrdU), which measures the incorporation of BrdU, a thymidine analogue, into the DNA of proliferating cells. The BrdU assay used in this study was an ELISA-based assay that was performed as recommended by the manufacturer (Merck-Millipore, USA). Imatimib treated K562 cells or ABL sgRNA virus infected K562 cells were incubated for 36 h at 37 °C, the media were supplemented with 10 μM BrdU and incubated for an additional 12 h. The cells were then stained with a peroxidase-labeled antibody against BrdU, followed by TMB Peroxidase Substrate addition for 30 min and acid stop solution exposure. The absorbance of the samples at 450 nm with a reference wavelength of 540 nm was measured using a microplate reader. 

### 2.3. Transfection and Cell Line Selection 

K562 cells were transfected with pcDNA3 plasmids expressing the firefly luciferase gene (the gene sequences were originally from *luc4.1*; Chris Contag, Stanford University, Stanford, CA, USA), as described previously [18]. Briefly, 1 × 10^6^ K562 cells were washed twice with phosphate-buffered saline (PBS) and mixed with 10 μg of plasmid. Two 1.2-kV pulses were applied for 20 milliseconds using a pipette-type Microporator MP-100 (Digital Bio, Seoul, Korea). Stable cells were selected 48 h later with G418 (1 mg/mL). Bioluminescent derivatives of K562 cells were used for further in vitro and in vivo studies.

### 2.4. Systemic Leukemia Animal Model

Four-week-old female severe combined immunodeficient (SCID) mice were purchased from the National Science Council Animal Center (Taipei, Taiwan) and housed in micro-isolator cages at the Laboratory Animal Service Center in the China Medical University (Taichun, Taiwan). This study was carried out in strict accordance with the recommendations in the Guide for the Care and Use of Laboratory Animals from the National Institutes of Health. The protocol was approved by the Institutional Animal Care and Use Committee (IACUC) at China Medical University (Permit Number: 2018-030-1). The systemic leukemia animal model uses immune-competent SCID mice to mimic cancer development in humans. Six mice were anesthetized with 2% isoflurane, and 5 × 10^6^ bioluminescent K562 cells were injected via the tail vein into each SCID mouse. The mice were then grouped (three mice/group) after three weeks, and injected via the tail vein with 5 × 10^8^ copy number of scramble (SC)- or *ABL*-targeting virus in a 20 μL normal saline. Throughout the study, all mice were kept in an environmentally controlled room maintained at 21–24 °C and 43–65% relative humidity. During the experiment, all animals underwent bioluminescence imaging every two weeks to observe the CML cells. All surgeries were performed under isoflurane anesthesia, and all efforts were made to minimize suffering. During the experiment, no stress or abnormal behavior due to the cancer was observed in the mice. The health status of the animals was monitored once daily by a qualified veterinarian.

### 2.5. Bioluminescence (IVIS) Imaging

Bioluminescence imaging was performed with a highly sensitive, cooled CCD camera mounted in a light-tight specimen box (In Vivo Imaging System-IVIS; Alameda, CA, USA). Fifteen minutes before imaging, the mice were injected i.p. with D-luciferin (200 mg/kg). The animals were placed on a warmed stage inside the camera box, and were continuously exposed to 2.5% isoflurane to sustain sedation during imaging. Every group of mice was imaged for 1, 5, 10, and 30 s. The light emitted from the mice was detected by the IVIS camera system, integrated, digitized and displayed. Regions of interest on the displayed images were identified, and the total photon counts were quantified using Living Image® software 4.0 (Caliper, Alameda, CA, USA). 

### 2.6. Flow Cytometry Analysis 

K562 cells (5 × 10^5^ cells/dish) were plated in 6-cm dishes for infection. An equal number of virus particles and K562 cells was defined as 1-fold. K562 cells were exposed to different folds of concentrated virus harboring the pLJM1-EGFP plasmid. Lentivirus-transduced cells were harvested three days after infection, and the GFP-positive cell population was analyzed by flow cytometry (FACSCalibur, BD Biosciences, San Diego, CA, USA).

### 2.7. Real-Time Quantitative Polymerase Chain Reaction (Q-PCR) 

Primers targeting the luciferase (forward 5′-CCGTCGTATTCGTGAGC-3′ and reverse 5′-GGTGGCAAATGGGAAGT-3′) and mouse β-glucuronidase (*GUS*, forward 5′-TGAACTCTTGAAAGCCTGC-3′ and reverse 5′-GAAATGGAGGACCAGCTCATA-3′) genes were used to quantify human CML DNA in the peripheral blood of the mice. Primers for the WPRE region (forward 5′-TCATGCTATTGCTTCCCGTA-3′ and reverse 5′-CCAAGGAAAGGACGATGAT-3′) were used for lentivirus quantification. All oligo primers were synthesized by Genomics BioSci and Tech (Taipei, Taiwan). A LightCycler thermocycler (Roche Molecular Biochemicals, Mannheim, Germany) was used for Q-PCR analysis. One microliter of sample and master mix was first denatured for 10 min at 95 °C and then subjected to 40 cycles (denaturation at 95 °C for 5 s; annealing at 60 °C for 5 s; and elongation at 72 °C for 10 s) with detection of fluorescence intensity. All the PCR samples underwent a melting curve analysis to detect non-specific PCR products. Luciferase gene expression from the Q-PCR analysis was normalized to mouse *GUS* expression as an indicator of DNA input using the built-in Roche LightCycler Software, version 4.

### 2.8. Absolute Q-PCR

To generate an absolute quantitative standard curve for Q-PCR analysis, we cloned the PCR product of the human *GUS* gene into the TA cloning vector (*pTA*® Easy Cloning Kit, Genomics BioSci and Tech, Taipei, Taiwan). After gene sequencing, *E. coli* amplification, plasmid purification and molecular weight determination, the copies of the *GUS* gene were calculated and diluted from 10^8^ to 10^2^ /μL. Each copied gene was measured for accuracy and a linear correlation.

### 2.9. Protein Extraction, Western Blotting, And Antibodies

For western blot analysis, K562 cells were washed once with ice-cold PBS and lysed with radioimmunoprecipitation assay (RIPA) lysis buffer containing protease inhibitors. Fifty micrograms of protein from each sample was resolved by sodium dodecyl sulfate polyacrylamide gel electrophoresis (SDS-PAGE) and transferred to a nitrocellulose membrane. The anti-GAPDH (sc-32233), anti-p-ERK (sc-7383), P21 (sc-817) and c-Abl (sc-23) antibodies were purchased from Santa Cruz Biotechnology (Santa Cruz, CA, USA), and the anti-PARP (#9541) antibodies were purchased from Cell Signaling Technology (Danvers, MA, USA). The secondary anti-mouse and anti-rabbit antibodies were purchased from Santa Cruz Biotechnology. Most of the primary antibodies were used at a 1:1000 dilution with overnight hybridization, whereas only the c-Abl antibody was used at a 1:250 dilution, followed by a one-hour incubation with a 1:4000 dilution of the secondary antibodies. All the western blotting was measured and quantified by Image J software. Original whole blot can be found at Appendix A. The comparison was done by fold of control cells. 

### 2.10. Plasmid Construction and Lentiviral Production

Lentiviral particles were produced by transient transfection of Phoenix-ECO cells (CRL-3214) using TransIT®-LT1 Reagent (Mirus Bio LLC, Madison, WI, USA). Guide oligonucleotides were phosphorylated, annealed, and cloned into the BsmBI site of the lentiCRISPR v2 vector (Addgene, 52961, kindly provided by Feng Zhang), according to the Zhang laboratory protocol [19] (F. Zhang lab, MIT, Cambridge, MA, USA). All the plasmid constructs were verified by sequencing. The lentiCRISPR construct or the pLJM1-EGFP plasmid (Addgene plasmid #19319, a gift from David Sabatini) was co-transfected with pMD2.G (Addgene plasmid #12259) and psPAX2 (Addgene plasmid #12260, both kindly provided by Didier Trono, EPFL, Lausanne, Switzerland). Lentiviral particles were collected at 36 and 72 h and then concentrated with a Lenti-X Concentrator® (Clontech, Mountain View, CA, USA). The lentivirus concentration for each gene was quantified by Q-PCR. Biohazards and restricted materials were used in this study in accordance with the “Safety Guidelines for Biosafety Level 1 to Level 3 Laboratory”. The protocol was approved by the Institutional Biosafety Committee (G-106-097) at Taipei Medical University, Taipei, Taiwan.

### 2.11. Design of on-Target and off-Target sgRNAs for the ABL and mABL Gene 

Custom sgRNAs for *ABL* and *mABL* gene were designed using the MIT CRISPR Design website (https://www.benchling.com/crispr/) with the sequence of *ABL* (NM_005157). This website provides both on-target sequences and off-target possibilities. We selected the highest scoring off-target sequences in the *ABL* protein-coding region, sgRNA_1 and sgRNA_2.

### 2.12. Sanger Sequencing and Gene Editing Efficiency Assay

Genomic DNA was extracted, and the *ABL* exon 2 region was PCR-amplified using the following primers: forward GAGAGGCTGGTGACACGTAA and reverse TTTGTAGAAAGCTTCCTTTTCCCG. Off-target sequences were PCR-amplified using the following primers: ADAMTSL1, forward TTTCTTCCTTTACTCTGCCAAATTA and reverse TACAATTCCAAGCTTCCGAT; TBRG4, forward TAGGGAGTAGATGCTCGTT and reverse GGACCTGGGAATCTGAATTAT; and C17orf75, forward CATGTCCCATCACTGCTC and reverse TTCTCCGTTTCATTCTGTGT. The *mABL* exon 2 region was PCR-amplified using the following primers: forward GGGAACCAAGTGAGACTATAC and reverse CAGGCATTTCTGCTCTCAA. The PCR products were purified using a PCR Clean-up Purification Kit and sequenced by Sanger sequencing using the forward PCR primers. The editing efficiency of the sgRNAs and the potential induced mutations were assessed using TIDE software (https://www.deskgen.com/landing/tide.html#/tide; Netherlands Cancer Institute, Amsterdam, Netherlands), which required only two Sanger sequencing runs from wild-type cells and mutated cells. 

### 2.13. RNA-Guided Engineered Nuclease-Restriction Fragment Length Polymorphism (RGEN-RFLP) Assay

PCR products (approximately 100 ng per assay) of the *ABL* exon 2 region were incubated for 30 min at 37 °C with Cas9 protein (30 nM) and sgRNAs (30 nM) in 10 μl of NEB buffer 3. After cleavage, RNase A (2 μg) was added, and the reaction mixture was incubated for 15 min at 37 °C to remove RNA. Next, proteinase K (2 μg) was added, and the reaction mixture was incubated for 15 min at 58 °C to remove the Cas9 protein. The products were resolved on 2% agarose gels and visualized by ethidium bromide (EtBr) staining.

### 2.14. Karyotype and Fluorescent In-Situ Hybridization (FISH) Analysis

Routine peripheral blood chromosomal analysis was performed using phytohemagglutinin (PHA) stimulation and standard techniques. Metaphase chromosomes were stained by giemsa-trypsin banding. Twenty metaphase cells were examined for each patient. The resolution of our protocol is around 500 bands on average. Fluorescence in situ hybridization was performed on interphase nuclei using Vysis LSI BCR/ABL dual color translocation probe set (Vysis, Downers Grove, IL, USA). Hybridization was carried out in a humidified chamber at 37 °C for at least 16 h. The slides were washed with 0.4× SSC/0.3% NP40 three times for 2 min each and then air-dried in the dark. Hybridization areas were counterstained with 20 μL DAPI (Vysis Inc, Downers Grove, IL, USA). Cells were observed under a Zeiss fluorescence microscope (Carl Zeiss Microimaging GmbH, Gottingen, Germany) and images were captured and analyzed using GenASIs Scan & Analysis platform of Applied Spectral Imaging (Carlsbad, CA, USA).

### 2.15. T-Cell Development and Lineage In Vivo

BALB/c mice were anesthetized with 2% isoflurane and tail-vein injected with 5 × 10^8^ copy number of scramble (SC)- or *mABL*-targeting virus in a 20 μL normal saline. After four weeks, the blood sampling was performed by cardiac puncture and stored in EDTA contained blood collection tube. The blood samples were then automatic hematology analyzed by IDEXX Procyte Dx (IDEXX Laboratories, Westbrook, ME, USA). The leukocytes from cardiac puncture collected blood, kidney, liver, lung and spleen from each mouse was isolated and Sanger sequenced for mouse *ABL* gene disruption analysis. 

### 2.16. Flow Cytometric Analysis and Cell Staining

Cardiac puncture collected blood were lysed of RBC and washed in FACS buffer (PBS containing 2% heat-inactivated FBS). For surface staining, the leukocytes were first incubated 5 min with anti-CD16/CD32 mouse Fc block (biolegend company, San Diego, CA, USA), followed by double staining with anti-CD4 (FITC) and CD8 (PE) antibodies (biolegend company, San Diego, CA, USA) for 30 min. Cells were washed twice, collected on a FACSCalibur flow cytometer, and analyzed using CellQuest software (BD Biosciences, San Diego, CA, USA). 

### 2.17. Statistical Methods

All data are expressed as the mean±standard error, and the differences were analyzed by Student’s t-test for pairwise samples. All statistical comparisons were performed using SigmaPlot graphing software (San Jose, CA, USA) and Statistical Package for the Social Sciences v.13 (SPSS, Chicago, IL, USA). A *p*-value < 0.05 was considered statistically significant, and all statistical tests were two-sided.

### 2.18. Ethics approval and consent to participate

This animal study was carried out in strict accordance with the recommendations of the *Guide for the Care and Use of Laboratory Animals* from the National Institutes of Health. The protocol was approved by the Institutional Animal Care and Use Committee (IACUC) at China Medical University (Permit Number: 2018-030-1).

The clinical patient study was approved by the Institutional Review Board (N201711069) in Taipei Medical University, Taipei, Taiwan, and the patients provided written consent.

## 3. Results

### 3.1. Optimization of Viral Transduction for Human K562 Cells

Several cell types, including primary human fibroblasts, human umbilical vein endothelial cells, Jurkat cells, and leukemia cells, have long been considered difficult to transfect [20]. To optimize leukemia cell transduction, we transduced CML (K562) cells with different concentrations of the EGFP expression virus (from 0.3 to 200:1; virus number:cell number). After transduction, the number of GFP-positive K562 cells was determined by flow cytometry (Appendix A); this number was significantly increased after infection at a ratio of 30:1 (purple), and reached a maximum at a ratio of 200:1 (yellow, Appendix A). We next examined the association between virus input and the GFP-positive K562 cell number. The results showed that a 100:1 virus ratio resulted in transduction of 40% of the K562 cell population, whereas additional virus did not significantly increase the transduction efficiency (defined as MOI = 1). In subsequent experiments, we used a MOI = 1 virus concentration for gene delivery (Appendix A).

### 3.2. Efficient and Specific CRISPR/Cas9 Gene Editing of the Human BCR-ABL Junctions in K562 CML Cells

For gene editing using CRISPR/Cas9 technology on *BCR-ABL* junctions, we amplified a DNA sequence of K562 CML cells using primers for the e12(b1) transcript on *BCR* and the a3 transcript on *ABL* [21]. The sequence showed *BCR*-*ABL* reciprocal translocation in K562 cells: sequence tags mapped to chr9:133,607,145-133,607,558 in the *ABL* gene were linked to chr22:23,632,242-23,632,742 in the *BCR* gene (Figure 1A), producing a P210 *BCR-ABL* Major form fusion protein [22]. Next, we designed two sgRNAs for the CRISPR/Cas editing system; protospacer 1 (*BCR*-*ABL*_1) targets the plus strand, whereas protospacer 2 (*BCR*-*ABL*_2) targets the negative strand (Figure 1B), specifically targeting the *BCR-ABL* junction sequence in K562 cells. Transduction of the K562 cells with the target scrambled (SC) virus produced a wild-type *ABL* sequence, as shown by Sanger sequencing (Figure 1C,D), with no evidence of gene editing. However, transduction with the *BCR*-*ABL* sgRNA_1 virus or the *BCR*-*ABL* sgRNA_2 virus led to multiple gene disruptions at the predicted cleavage sites (red arrowhead, Figure 1E,F) of both protospacers. In addition, both *BCR*-*ABL* sgRNA_1 and *BCR*-sgRNA_2 virus infections showed considerable gene editing efficiency, with 53% and 97.5% editing of the cell pools shown by TIDE analysis, respectively (Figure 1G,H, Appendix A). The most frequent mutations in the *BCR*-*ABL* sgRNA_1 cell pool were other edits (37.6%) and 2-bp deletions (10.1%), whereas those in the *BCR*-*ABL* sgRNA_2 cell pool were other mutations (77.3%) and 1-bp insertions (11.6%). The algorithm predicted the same patterns of genome repair for *BCR-ABL* sgRNA_1 and *BCR-ABL* sgRNA_2, which included mutations mainly at the cleavage sites (Appendix A). Next, we determined whether cancer cell viability is influenced by *BCR-ABL* gene disruption in K562 cells. We then confirmed the gene editing efficiency by assessing ABL protein expression. We observed that both the 145-kDa ABL protein and the 210-kDa BCR-ABL oncoprotein present in K562 cells (Figure 1I). Upon infection with the *BCR-ABL* sgRNA_1 and *BCR-ABL* sgRNA_2 virus, the BCR-ABL oncoprotein levels were significantly decreased, compared to control and scramble virus-infected cells. Protein analysis also demonstrated that PARP cleavage and the tumor suppressor P21 were both significantly induced in *BCR-ABL* sgRNA virus-infected K562 cells. We also determined whether cancer cell viability is influenced by *BCR-ABL* gene disruption in K562 cells. The results showe that the viability of K562 cells was significantly inhibited by infection with the *BCR-ABL* sgRNA_2 virus (*p* < 0.001) compared to infection with the *BCR-ABL* sgRNA_1 virus (*p* = 0.011) and scramble virus (Figure 2J). These findings demonstrated that targeting the *BCR-ABL* gene caused a dramatic disruption in DNA sequence and protein suppression and eventually suppressed cell viability in CML cells, implying that *BCR-ABL* gene editing would be a sufficient and effective anticancer strategy. However, in clinical patients, the genomic breakpoints in both *BCR* and *ABL* of CML cells are dispersed over intervals of 3.0 kb and ~150 kb, respectively. Each patient’s fusion sequence is therefore virtually unique, indicating that a patient-specific *BCR-ABL* CRISPR/Cas9 gene editing strategy for anticancer therapy will be a challenge. To be more specific, due to the limited option on *BCR-ABL* junction sequence in CML patients, it will be difficult to design specific and effective sgRNAs with appropriate PAM (NGG) sequence for CRISPR/Cas9 based personalized medicine. Accordingly, such sites only occur once in every 128 bp of random DNA sequence [23].

### 3.3. CRISPR/Cas9 Sufficiently Disrupts the ABL Gene in K562 Cells

To develop a novel CML therapy for clinical use, we utilized CRISPR/Cas9 genomic editing by targeting two custom-designed protospacers on the human *ABL* locus. As shown in the *ABL* genomic map on chromosome 9 (Figure 2A), protospacer 1 (*ABL* sgRNA_1) targets the plus strand, whereas protospacer 2 (*ABL* sgRNA_2) targets the negative strand. Transduction of K562 cells with the target scrambled (SC) virus produced a wild-type *ABL* sequence, as shown by Sanger sequencing (Figure 2B,C), with no evidence of gene editing. However, transduction with the *ABL* sgRNA_1 virus or the *ABL* sgRNA_2 virus led to multiple gene disruptions at the predicted cleavage sites (Figure 2D,E). In addition, *ABL* sgRNA_2 virus infection was shown by TIDE analysis to have strong gene editing efficiency: 41.2% of the cell pool was edited (Figure 2G,I) compared to only 8.8% with the *ABL* sgRNA_1 virus (Figure 2F,H). The most frequent mutations in the *ABL* sgRNA_1 cell pool were 1-bp insertions (3.4%) and 1-bp deletions (3.1%), whereas those in the *ABL* sgRNA_2 cell pool were 1-bp insertions (38.4%) and other mutations (2.7%). Again, the algorithm predicted the same patterns of genome repair for both *ABL* sgRNAs, which included mutations mainly at the cleavage sites (Appendix A).

### 3.4. ABL Loss Significantly Inhibits K562 Cancer Cell Growth and Induces Apoptosis

To confirm the gene editing efficiency of both *ABL* sgRNAs, we assessed the PCR amplification of the gene editing site by RNA-guided engineered nuclease-restriction fragment length polymorphism (RGEN-RFLP) analysis. To perform this assay, we first purified the Cas9 protein by affinity chromatography pull-down of the histidine tag in *E. coli* lysate containing NLS-Cas9 (Appendix A). The molecular weight of the purified Cas9 protein was approximately 140 kDa. RGEN-RFLP analysis of the gene editing efficiency of the *ABL* DNA region showed that the SC sgRNA without Cas9 did not cleave the DNA (Figure 3A); however, the SC sgRNA with Cas9 fully cleaved the DNA into fragments, indicating a 100% wild-type DNA sequence without gene disruption. In contrast, the proportion of uncut DNA from K562 cells was higher after exposure to *ABL* sgRNA_2 than *ABL* sgRNA_1, indicating that *ABL* sgRNA_2 had a higher gene editing efficiency than *ABL* sgRNA_1, with 38% versus 6.8% indels in the *ABL* gene region (the fragment from the DNA cleavage is indicated with an asterisk). To confirm the gene editing efficiency, we assessed ABL protein expression by western blot analysis. We observed that both the 145-kDa ABL protein and the 210-kDa BCR-ABL oncoprotein are present in K562 cells (Figure 3B). Upon infection with the *ABL* sgRNA_2 virus, the BCR-ABL oncoprotein levels were significantly decreased, compared to those in the *ABL* sgRNA_2 virus- and scramble virus-infected cells. Protein analysis also demonstrated that the tumor suppressor P21 was significantly induced in *ABL* sgRNA_2 virus-infected K562 cells. This observation directly explains the strong induction of apoptosis caused by PARP cleavage in the *ABL-*edited leukemia cells. Next, we determined whether cancer cell viability is influenced by *ABL* gene disruption in K562 cells. The results showed that the viability of K562 cells was significantly inhibited by infection with the *ABL* sgRNA_2 virus (*p* = 0.001) compared to infection with the *ABL* sgRNA_1 virus and scramble virus (Figure 3C). In addition, the cell proliferation of K562 cells shown by BrdU incorporation was significantly inhibited by infection with the *ABL* sgRNA_2 virus (*p* < 0.05), compared to that of the SC sgRNA-infected cells (Figure 3D). These findings demonstrated that the targeting *ABL* gene have similar anti-cancer effect of targeting *BCR-ABL* gene, causing a dramatic suppression of both cell viability and cell proliferation in CML cells. Notably, in difficult-to-transfect cells, a high level of gene editing is possible with a virus with an optimized sgRNA, indicating the future potential for gene therapy in all CML patients.

### 3.5. In Vivo ABL-Targeted Gene Editing Effectively Inhibits Leukemia Cell Growth

To evaluate the therapeutic effects of *ABL* disruption in vivo, we established a systemic leukemia animal model to evaluate the anticancer effect of an *ABL*-targeted CRISPR/Cas9. Luciferase-labeled human CML K562 cells were purified and expanded under normal culture conditions. SCID mice were injected with luciferase-labeled K562 cells through the tail vein. After three weeks, the mice were grouped and then injected via the tail vein with the SC- or *ABL* sgRNA_2-targeted CRISPR/Cas9 virus. The bioluminescence in each mouse was detected ventrally (Figure 4A) and dorsally (Figure 4B) every two weeks, to evaluate changes in leukemia cell number. The bioluminescence images clearly showed strong K562 cell growth in animals injected with the SC-targeted CRISPR/Cas9 virus from week 3 to week 7, especially in the ventral images of the heart and brain (blood-enriched organs). However, in mice treated with the *ABL*-targeted CRISPR/Cas9 virus, the K562 cell number only slightly increased in the ventral and dorsal bioluminescence analyses. Radiance photon measurements were used to calculate the cell number in each mouse, and the K562 cell growth curve showed significantly greater growth in the SC group than in the *ABL*-targeted group in both the ventral (*p* = 0.02, Figure 4C) and dorsal (*p* = 0.03, Figure 4D) analyses. Notably, this leukemia animal model not only mimics a gene therapy application in CML patients but also indicates that one course of *ABL*-targeted CRISPR/Cas9 virus treatment is sufficiently effective to suppress leukemia cell growth. To confirm the bioluminescence observations, we stained peripheral blood smears from the SC and *ABL* groups (Figure 4E). Myoblasts and neutrophils are shown at 400× and 1000× magnification; the smears showed a significant difference in the immature differential white blood cell count in the SC and *ABL* groups, with average values of 35 and 9, respectively (*p* < 0.05, Figure 4F). In addition, enhanced platelet aggregation and an increased basophil number were found in the SC group, mirroring the clinical diagnostic features of CML patients. Finally, we purified DNA from the peripheral blood of mice in the SC and *ABL* groups. Absolute quantification of the DNA was performed with luciferase-specific primers, and the results were normalized to the mouse *GUS* gene copy number in the qPCR analysis (Figure 4G). Again, the data showed significantly higher luciferase gene (carried by the CML cells) expression in the SC-targeted CRISPR/Cas9 virus-treated mice than in the *ABL*-targeted CRISPR/Cas9 virus-treated mice, indicating a strong therapeutic effect of *ABL*-targeted gene therapy in a systemic animal model of human leukemia.

### 3.6. Ex Vivo ABL-Targeted Gene Editing of Clinical CML Patients

To evaluate whether ABL-targeted gene editing can be used as an antileukemia therapy in the clinic, we infected the peripheral blood mononuclear cells (PBMCs) of CML patients with the *ABL* sgRNA_2-targeted CRISPR/Cas9 virus. The patient was a 49-year-old male who had lost 10 kg of body weight within one year, and complained of a low-grade fever, early satiety and loss of energy. The laboratory examination of this patient showed normal values for GOT, GPT, creatinine, AFP, CEA and CA199. However, the hematology analysis showed that RBC, lymphocyte and monocyte counts were far below average, with poor RBC quality, as shown in the HGB and HCT measurements. The WBC count was 320.8 × 10^3^/μL (normal range is 4.8–10.8 × 10^3^/μL), and the WBC differential count identified 12% band cells, 8% metamyelocytes, 4% promyelocytes and 5% blasts of all immature leukocyte stages in peripheral blood. Conventional cytogenetic analysis of products of conception revealed, at a 400 G-band level of resolution, an abnormal karyotype 46, XY, t(9;22)(q34;q11.2) (Figure 5A). Clear karyotypes of chromosome 9 and chromosome 22 are also shown (Figure 5B). To further confirm the chromosomal translocation, we performed a FISH analysis of the patient through the hybridization of combined probes targeting chromosome 9 and centromere 22 (Figure 5C). Our data revealed that the “fusion” signal resided on the derived chromosome 9 and not on the derived chromosome 22, as would typically be expected in CML with one isolated fusion signal. Next, we investigated the efficacy of CRISPR/Cas9 genome editing in clinical CML cells by SC, *ABL* sgRNA_1 and *ABL* sgRNA_2 virus infection with the optimized virus concentration. The CML cells transduced with or without the SC virus produced a wild-type *ABL* sequence, as shown by Sanger sequencing (Figure 5D,E), with no evidence of gene editing. However, transduction with the *ABL* sgRNA_1 or *ABL* sgRNA_2 virus led to multiple gene disruptions at the predicted cleavage sites (red arrowhead) of both protospacers, with more pronounced effects in *ABL* sgRNA_2-transduced CML cells than *ABL* sgRNA_1-transduced CML cells (Figure 5F,G). In addition, *ABL* sgRNA_2 virus infection was shown to have strong gene editing efficiency by TIDE analysis: 30.9% of the cell pool was edited, rather than 9.4% with a 1-bp deletion in *ABL* sgRNA_1 virus infection (Figure 5H–J). The most frequent mutations in the *ABL* sgRNA_2 CML cell pool were 1-bp insertions (27.7%) and other mutations (1.9%) (Figure 5K). Next, we used a LIVE/DEAD assay to visualize apoptosis induced by CRISPR/Cas9 genome editing of *ABL* in clinical CML cells (Figure 5L). The images showed a significant increase in the number of cells undergoing apoptosis after transduction with *ABL* sgRNA_2 (21.3%±4.65, apoptosis cell percentage), compared to that of SC (2.1% ± 1.2, apoptosis cell percentage)-transduced CML cells (Figure 5M, *p* < 0.05), indicating that the *ABL*-targeted CRISPR/Cas9 virus can be used an effective gene therapy strategy for clinical CML patients.

### 3.7. High Specificity of the ABL-Targeted CRISPR/Cas9 System Without Potential off-Target Effects

As highly efficient CRISPR/Cas9 gene editing technology has been used in vitro, in vivo and ex vivo, it is critical to determine whether this technology causes unexpected cleavage events at similar DNA sequences or under other circumstances. If so, further analysis of the specificity of CRISPR/Cas9 technology should be undertaken. From all human genes, we chose those with highly similar sequences to the *ABL* sgRNA_1 and *ABL* sgRNA_2 sequences as candidates for off-target cleavage and evaluated these genes by Sanger sequencing. The genes with a highly similar DNA sequence to the *ABL* sgRNA_1 sequence, which might thus be subject to off-target effects, are ADAMTSL1 and TBRG4 (Figure 6A), and the C17orf75 gene shares sequence similarity with *ABL* sgRNA_2. We designed specific primers for these three genes, amplified the regions by PCR, and analyzed the DNA sequences by Sanger sequencing. No genomic editing occurred in these genes (Figure 6B). This result validates the high specificity of the CRISPR/Cas9 system in targeting the *ABL* gene, and the superior leukemia therapeutic effects observed both in vitro and in an animal model are anticipated to translate into the clinic in the future.

### 3.8. The ABL Gene Editing Showed Better Anticancer Effects than Imatinib Treatment in Imatinib-Resistant K562 Cells

Imatinib (Gleevec) is a chemotherapeutic used to treat Ph-positive CML and ALL and certain types of gastrointestinal stromal tumors, systemic mastocytosis and myelodysplastic syndrome [24]. To investigate the anticancer activity of imatinib and the essential role of ABL in K562 cells in vitro, we determined the IC50 value of imatinib in K562 cells by treating them with different concentrations of imatinib for 48 h and then performing MTT assays (Figure 7A). K562 cells were highly sensitive to imatinib (IC50 = 1.8 μM). In addition, imatinib inhibited K562 cell viability, perhaps by suppressing ERK activation and upregulating the tumor suppressors P21 and P27, eventually resulting in PARP cleavage and apoptosis (Figure 7B). Next, we used a LIVE/DEAD assay to visualize imatinib-induced apoptosis (Figure 7C). The images showed an increasing number of K562 cells undergoing apoptosis with increasing concentrations of imatinib, with significant cell death at 10 and 25 μM imatinib compared to the control (Figure 7D, *p* < 0.05). In addition, at these concentrations, imatinib significantly inhibited K562 cell viability and cell proliferation by approximately 50% compared to the DMSO control, whereas only sgRNA_2 virus infection had a similar inhibitory effect on K562 cell viability (Figure 7E, *p* < 0.05). Our recent study showed that imatinib-resistant K562 cells (K562-IR) have a much higher IC50 value for imatinib treatment than wild-type K562 cells, with a 30-fold increase in drug sensitivity [25]. With the same cells, we did not find any mutations in the entire *ABL* gene through Sanger sequencing, indicating that BCR-ABL amino acid substitutions inside the kinase domain may not be the main cause of disruption in the interaction of imatinib and the tyrosine kinase domain, resulting in a loss of sensitivity to the drug in our long-term imatinib-treated K562-IR cells. With imatinib treatment and *ABL*-edited virus infection of K562-IR cells, we found that K562-IR cells with the *ABL* sgRNA_2 virus showed significant cancer cell growth inhibition (Figure 7F, *p* < 0.05), which was even higher than that with the high concentration (25 μM) of imatinib. Finally, we confirmed that both *ABL sgRNA_1* and *ABL sgRNA_2* virus introduction into the normal bone marrow cell line HS27A resulted in no difference in cell growth compared to that of SC virus-infected cells (Figure 7G), ensuring the safety of the *ABL* target virus in vivo. The above evidence indicates that *ABL* plays an important role in promoting cell growth and preventing apoptosis. CRISPR/Cas9-based gene editing of *ABL* provides an effective anticancer strategy for imatinib-resistant CML cells.

### 3.9. Mouse ABL Editing through CRISPR/Cas9 to Investigate T-Cell Survival and Development In Vivo

Previous studies showed that *ABL* gene has its own physiological roles such as regulating T-cell survival and development [26,27], which is import for anti-cancer immune response. To investigate this potential concern, we targeted *ABL* gene on mouse chromosome 2 to investigate whether *ABL* disruption through CRISPR/Cas9 affects T-cell survival and development in vivo (Figure 8A). We utilized CRISPR/Cas9 genomic editing by targeting two mouse protospacers as protospacer 1 (*mABL* sgRNA_1) targets the plus strand, whereas protospacer 2 (*mABL* sgRNA_2) targets the negative strand. Transduction of NIH-3T3 mouse fibroblast cells with the target scrambled (SC) virus produced a wild-type *ABL* sequence, as shown by Sanger sequencing (Figure 8B,C), with no evidence of gene editing. However, transduction with the *mABL* sgRNA_1 virus and the *mABL* sgRNA_2 virus transfection cells led to major single nucleotide deletions around the predicted cleavage sites (Figure 8D,E). Through TIDE analysis, it is showed that both *mABL* sgRNA_1 and *mABL* sgRNA_2 obtained great gene editing efficiencies with 97.4% (Figure 8H) and 98.9% (Figure 8I) of the cell pool, respectively. The most frequent mutations in the *mABL* sgRNA_1 cell and *mABL* sgRNA_2 cell pool were 1-bp delection with 86.3% (Figure 8F) and 90.1% (Figure 8G), respectively. In addition, the algorithm predicted the same patterns of genome repair for both *mABL* sgRNAs, which included mutations mainly at the cleavage sites (Appendix A). Next, we tail-vein injected SC-, *mABL* sgRNA_1- and *mABL* sgRNA_2 virus on BALB/c mice. After four weeks, the mice were scarified and the blood samples were collected and hematology analyzed, whereas isolated leukocytes and internal organs were Sanger sequenced for mouse *ABL* gene disruption. The representing result showed that the *mABL* sgRNA_1 and *mABL* sgRNA_2 virus transfected mice obtained great *ABL* gene editing on leukocytes (Appendix A, Appendix A), with 55 ± 5.29% and 64.66 ± 7.13% of all leukocyte pool, respectively (Figure 8J). Interestingly, all the *ABL* gene sequences from internal organs, such as the kidney, liver, lung and spleen remained unedited, either in SC- or *mABL* sgRNAs introduced mice (Appendix A). This observation suggests that lentivirus as a gene carrier has a short-term but great transfection effect on the circulation cells through tail-vein delivery, this could prevent unwanted host targeting. In Table 1, the hematology analysis of RBC, WBC and platelet cell counts showed no significant differences between SC-, *mABL* sgRNA_1- and *mABL* sgRNA_2 targeted CRISPR/Cas9 virus injected mice. Furthermore, the percentage of neutrophil, monocyte and lymphocyte from DIFF scattergram showed no differences among these three groups (Appendix A), indicating mouse *ABL* gene editing through CRISPR/Cas9 has no significant effect on either blood cell survival or differentiation. To further reveal whether mouse *ABL* gene editing influences T-cell development, we used flow cytometry to determine CD4 and CD8 expressions of blood purified leukocytes from three groups of mice. Using unstained samples to determine CD4 and CD8 positive regions (CD4^+^ and CD8^+^), it is clear shown that the percentages of CD4^+^CD8^+^DP (double positive), CD4^-^CD8^-^DN (double negative) and CD4^+^ or CD8^+^ SP (single positive) T-cells were not significantly different between SC and *ABL* gene edited mice (Figure 8K). These results suggested that T-cell linage of CD4^+^ or CD8^+^ lymphocytes are not effected in the absence of mouse ABL gene.

### 3.10. Summary of the Research Strategy in this Study

In this study, we aimed to investigate the anticancer efficacy of virus-mediated CRISPR/Cas9 gene therapy targeting the human *ABL* gene in CML cells in vitro and in vivo (Figure 8). First, the *ABL*-targeting virus was carefully produced, quantified and optimized for CML cell delivery (Figure 9A). The concentrated *ABL*-targeting virus was used both in vitro (Figure 9B) and in an in vivo bioluminescence imaging-based systemic leukemia animal model (Figure 9C); mouse systemic leukemia can be considered a preclinical gene therapy model (Figure 9D). Once CML cells are infected with the *ABL*-targeting virus (Figure 9E), the sgRNA, with the PAM sequence, targets the *ABL* gene locus (Figure 9F), and the sgRNA loop then carries Cas9 to cleave the nearby DNA. The competing NHEJ pathway for DNA repair is often favored and frequently leads to indels or chromosomal rearrangements, particularly in mammalian cells (Figure 9G). These DNA indels result in frameshifts in the protein-coding sequence, thus producing nonsense proteins or causing early termination of ABL translation, eventually resulting in CML cell death (Figure 9H).

## 4. Discussion

In the past few years, CRISPR/Cas9 gene editing technology has become an important strategy for discovering therapeutic targets in human disease. In basic laboratories, most studies continue to use plasmid transfection approaches, such as liposomes or electroporation, to deliver CRISPR/Cas9 and sgRNA for gene editing. One major limitation of those methodologies is that they may alter the physical or biological condition of the target cells, or even cause incidental cell death unrelated to gene targeting. Here, we used lentiviruses as carriers to deliver the CRISPR/Cas9 genomic editing system to achieve improved transfection efficiency and increasingly flexible treatment applications for both in vitro and in vivo experimental designs. Lentivirus use has grown exponentially both in research and in gene therapy protocols, accounting for 12% of the viral vector-based clinical trials in 2011, and the percentage is still increasing [28]. However, the safety of lentivirus use should be the highest priority in the clinic, and many questions remain about the potential harm that lentiviruses may cause. Thus, it will be difficult to balance the inherent potential risks with the potential benefits of abrogating oncogenes with this powerful tool. Therefore, the appropriate lentiviral input should be determined in an animal model of systemic leukemia to avoid unexpected side effects.

Ideally, genomic editing will be optimized to enhance on-target efficiency and to reduce off-target efficiency. However, off-target cleavage and other undesirable effects on protein translocation and recombination have considerable potential to hinder CRISPR/Cas9 research and related biotechnology [29]. Off-target effects stem from nonspecific recognition of non-target sequences, which might lead to alterations in protein expression. The major concern is that gene editing-mediated chromosomal rearrangements might disable tumor suppressor genes or activate oncogenes, both of which could contribute to cellular toxicity [30]. Several techniques to measure off-target effects, such as targeted sequencing, exome sequencing, whole genome sequencing and GUIDE-sequencing, are often used. Each method has its own advantages and disadvantages. Targeted sequencing is relatively easy, fast and widely available, although it presents some potential drawbacks. For example, the results of targeted sequencing might be biased if no unexpected mutation sites are detected. Off-target measurements might be expensive and time-consuming if many candidate sites are screened. In this study, we used Sanger sequencing (targeted sequencing) to demonstrate that our developed *ABL*-targeted therapy had no off-target effects on similar gene sequences in the pool of gene-edited cells, indicating that the *ABL*-based CRISPR/Cas9 gene editing system is highly specific and would be safe for CML therapy. However, more sensitive off-target detection methods are required, especially for applications such as gene therapy that require absolute fidelity. Further research should use unbiased GUIDE-sequencing (next-generation sequencing) to ensure off-target detection with high sensitivity and high DNA coverage before applying *ABL*-based CRISPR/Cas9 gene editing in a clinical setting.

Due to the success of CRISPR/Cas9 genomic editing, many studies have used this technology to target *BCR-ABL* fusion genes in different models for different purposes. For example, Lekomtsev and colleagues employed Cas9 with two guide RNAs targeting the two breakpoints in the haploid human cell line eHAP, to determine whether *BCR-ABL* translocations could be reverted back to the wild-type status. However, after genotyping analysis showed that the overall targeting efficiency of the *BCR-ABL* translocations was 0.8% of 384 clones, additional spectral karyotyping and G-band karyotyping were used to confirm and visualize the engineered chromosomal translocations [31]. Additionally, Hara and his colleagues investigated the efficiency of both on-target and off-target genome editing by introducing FokI-dCas9 (fCas9), Cas9 D10A (D10A), Cas9 WT (WT) and gRNAs into zygotes to generate mutant mice [32]. The results of their study revealed that the specificity of fCas9 is more strictly regulated than that of other Cas9 forms, enabling the generation of knockout mice with reduced unwanted off-target effects by CRISPR/Cas9 technology.

In another study using CRISPR/Cas9 genomic editing of the *BCR-ABL* fusion gene, García-Tuñón and his group were the first to report the use of CRISPR/Cas9 genome editing to abrogate the human BCR-ABL oncoprotein in leukemia cells as a therapeutic intervention [33]. The authors designed specific sgRNAs to direct Cas9 to the *BCR/ABL* fusion sequence (junction sequence) in the Boff-p210 cell line, a pro-B-derived Baf/3 hematopoietic cell line that artificially expresses BCR/ABL. The CRISPR/Cas9-mediated reduction in BCR-ABL oncoprotein (p210) expression in Boff-p210 cells resulted in the loss of tumorigenicity in a CML xenograft animal model. However, several forms of *BCR-ABL* fusion proteins have been identified in the clinic based on three breakpoint regions—major breakpoint cluster region (M-BCR, P210, 210 kDa), minor breakpoint cluster region (m-BCR, P190, 190 kDa) and micro breakpoint cluster region (mu-BCR, P230, 230 kDa)—or, in rare cases, other nearby sites [3,34]. Thus, targeting the junction sequences of *BCR/ABL* may not be applicable to all clinical CML patients. Therefore, in this study, we utilized a methodology similar to that in previous work. Our study adopted approaches like those used for clinical gene therapy, such as using lentiviruses to deliver CRISPR/Cas9 genes, targeting leukemia cells with the native Ph chromosome, optimizing the *ABL* genomic editing sequences and evaluating cancer efficacy in an animal model of systemic leukemia. Notably, our study produced a more reliable and flexible *ABL*-based gene therapy than approaches targeting the *BCR-ABL* oncogene junction in CML patients. However, although any adverse effect was not observed with CRISPR/Cas9-directed disruption of the *ABL* gene in mice in this short-term study, it should be noted that the long-term effect of ABL gene disruption is not known. Furthermore, the extent to which *ABL* gene disruption affects the hematopoietic stem cell population is not yet known. In the future, the genomic editing of *ABL* described in this study could be improved by addressing the following concerns: (a) primary cultures of leukemia cells from CML patients are required to assess anticancer efficacy; (b) lentivirus transfection may vary between primary leukemia cells and the K562 cell line; (c) a primary culture immortalization system for leukemia cells is essential for future preclinical anticancer evaluation; (d) a systemic leukemia model using primary culture cells would be the best platform to mimic a gene therapy approach targeting *ABL* or other oncogenes. Nonetheless, to the best of our knowledge, this is the most relevant preclinical study of *BCR-ABL*-targeted therapy, the efficacy of which stems from virus-mediated genome editing. Such findings are useful for future studies and the optimization of gene therapy in clinical trials.

## 5. Conclusions

In conclusion, this study established a virus-mediated *ABL*-targeting gene therapy that significantly reduced Ph expression and abolished leukemia cell survival and tumorigenic abilities in vivo, in vitro and ex vivo. These findings suggest that this CRISPR-Cas9-based gene therapy has strong potential and may be further applied for the treatment of CML patients who are insensitive or resistant to imatinib treatment.

## Figures and Tables

**Figure 1 cancers-12-01399-f001:**
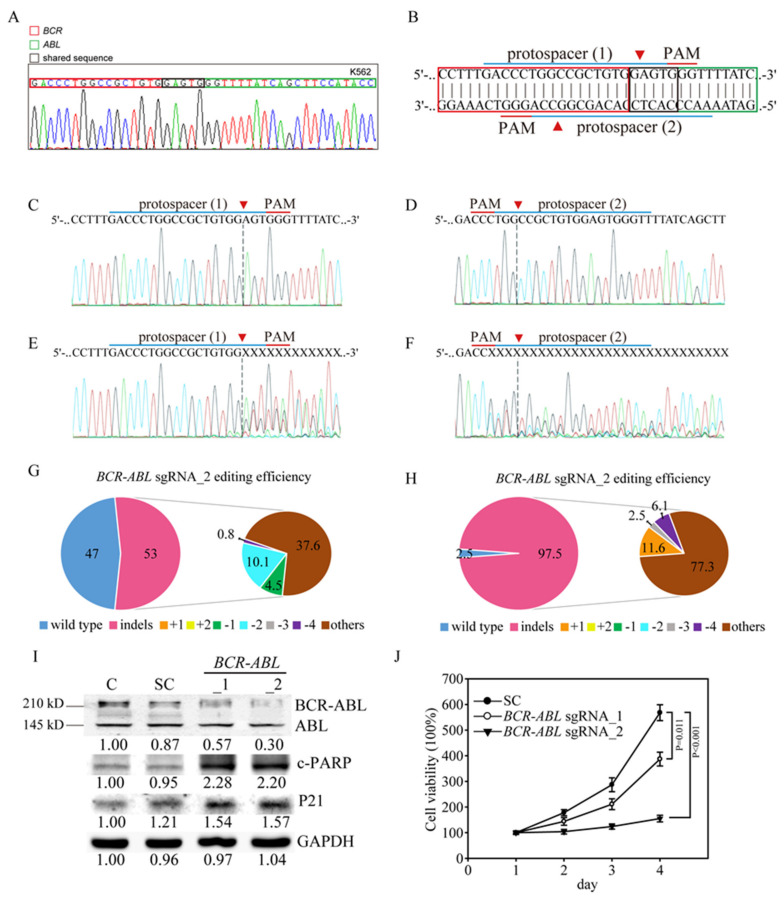
Efficient and specific clustered regularly interspaced short palindromic repeats (CRISPR/Cas9) gene editing of the *BCR-ABL* junction regions in K562 cells. (**A**) DNA sequence map of the *BCR-ABL* junctions in K562 cells. The red column indicates the *BCR* gene, the red column indicates the *ABL* gene and the black column indicates the shared sequences for both the *BCR* and *ABL* genes. (**B**) Schematic representation of the human *BCR-ABL* junctions and two protospacer sequences (blue underline) for editing that were designed from the plus (protospacer 1) and negative (protospacer 2) DNA strands. The arrowhead indicates the expected Cas9 cleavage site. The protospacer adjacent motif (PAM, red underline) is the motif required for Cas9 nuclease activity. Scrambled (SC) sgRNA and *BCR-ABL* sgRNA were delivered to K562 cells by lentivirus. After transduction, DNA from the virus-infected cells was purified and subjected to Sanger sequencing of (**C**) plus and (**D**) negative DNA strands of the *BCR-ABL* junction in the SC K562 cells. (**E**) *BCR-ABL* sgRNA_1 and (**F**) *BCR-ABL* sgRNA_2 produced a mixture of sequences around the expected Cas9 cleavage site in a pool of gene-edited cells after lentivirus transduction. Tracking of indels by decomposition (TIDE) algorithm analysis of the *ABL* gene-edited sequences (indels, insertions and deletions) showed a high editing efficiency in K562 cells. The pie charts show the percentages of indels in the *ABL* gene edited by (**G**) *BCR-ABL* sgRNA_1 and (**H**) *BCR-ABL* sgRNA_2. The gene editing efficiency of the two sgRNAs is presented in red, while the most common other indels is presented in brown color. (**I**) Western blot analysis of BCR-ABL and ABL protein expression. Bands at 210 kDa and 145 kDa, corresponding to BCR-ABL and ABL, respectively, were observed in the parental and scrambled (SC) sgRNA control K562 cells, whereas these bands were significantly reduced in the *BCR-ABL* sgRNA_1- and *BCR-ABL* sgRNA_2-transduced K562 cells. The protein expression of downstream ABL targets, such as P21 and cleaved PARP (c-PARP), was also activated by *BCR-ABL* gene redundancy. All the western blotting was measured and quantified by Image J software. (**J**) Cell viability curve of the *BCR-ABL* sgRNA_1- and *BCR-ABL* sgRNA_2-transduced K562 cells determined by 3-(4,5-dimethylthiazol-2-yl)-2,5-diphenyltetrazolium (MTT) assays. Data are presented as the mean and standard deviation. Data were analyzed with Student’s *t*-test.

**Figure 2 cancers-12-01399-f002:**
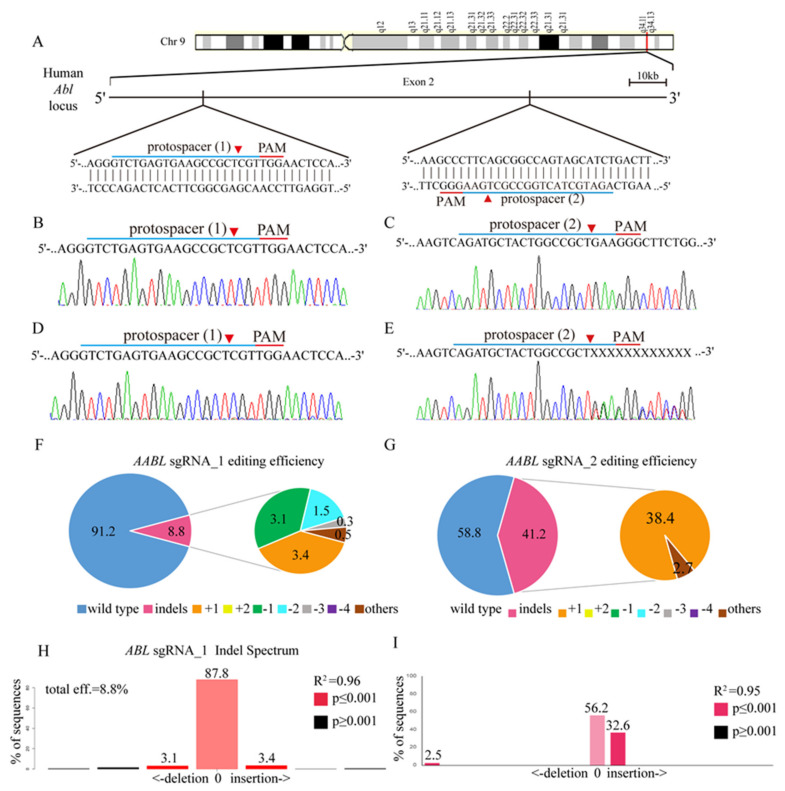
*ABL* gene targeting in K562 cells using the CRISPR/CAS9 system. (**A**) Schematic representation of the human *ABL* DNA locus and two protospacer sequences (blue underline) for editing. The arrowhead indicates the expected Cas9 cleavage site. The protospacer adjacent motif (PAM, red underline) is the motif required for Cas9 nuclease activity. Scrambled (SC) sgRNA and *ABL* sgRNA were delivered to K562 cells by lentivirus. (**B**,**C**) After transduction, DNA from the virus-infected cells was purified and subjected to Sanger sequencing of the wild-type *ABL* DNA locus in K562 cells. (**D**) *ABL* sgRNA_1 and (**E**) *ABL* sgRNA_2 produced a mixture of sequences around the expected Cas9 cleavage site in a pool of gene-edited cells after lentivirus transduction. TIDE algorithm analysis of the *ABL* gene-edited sequence (indels, insertions and deletions) showed a high editing efficiency in K562 cells. The pie charts show the percentages of indels in the *ABL* gene edited by (**F**) *ABL* sgRNA_1 and (**G**) *ABL* sgRNA_2. The gene editing efficiency of the two sgRNAs is presented in red, while the two most common -1 and +1 indels are presented in green and orange, respectively. The TIDE analysis of indel distribution is shown for (**H**) the *ABL* sgRNA_1- and (**I**) *ABL* sgRNA_2 virus-transfected K562 cells compared to the SC K562 cells.

**Figure 3 cancers-12-01399-f003:**
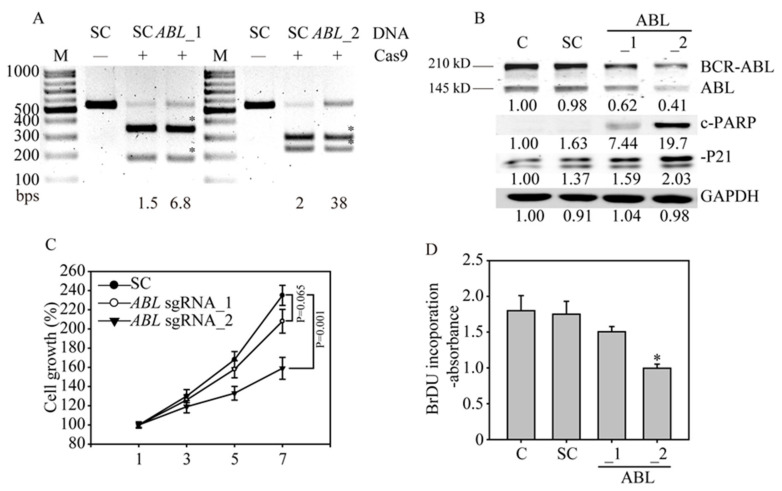
*ABL* gene disruption inhibits cancer cell growth and induces apoptosis in K562 cells. (**A**) The *ABL* gene in K562 cells was analyzed with RNA-guided engineered nuclease-restriction fragment length polymorphism (RGEN-RFLP) assays to measure the gene editing efficiency. The gel image of *ABL* gene cleavage induced by addition of specific sgRNA, and Cas9 shows the indel percentage in the gene editing pool. Cleaved DNA fragments are highlighted with an asterisk. (**B**) Western blot analysis of BCR-ABL and ABL protein expression. Bands at 210 kDa and 145 kDa, corresponding to BCR-ABL and ABL, were observed in the parental and SC sgRNA control K562 cells, whereas these bands were significantly reduced in the *ABL* sgRNA_1- and *ABL* sgRNA_2-transduced K562 cells. The protein expression of downstream ABL targets, such as P21 and cleaved PARP (c-PARP), was also activated by *ABL* gene redundancy. All the western blotting was measured and quantified by Image J software. (**C**) Cell viability curve of the *ABL* sgRNA_1- and *ABL* sgRNA_2-transduced K562 cells determined by MTT assays. (**D**) Cell proliferation of the *ABL* sgRNA-transduced K562 cells was determined by bromodeoxyuridine (BrdU) incorporation assays.

**Figure 4 cancers-12-01399-f004:**
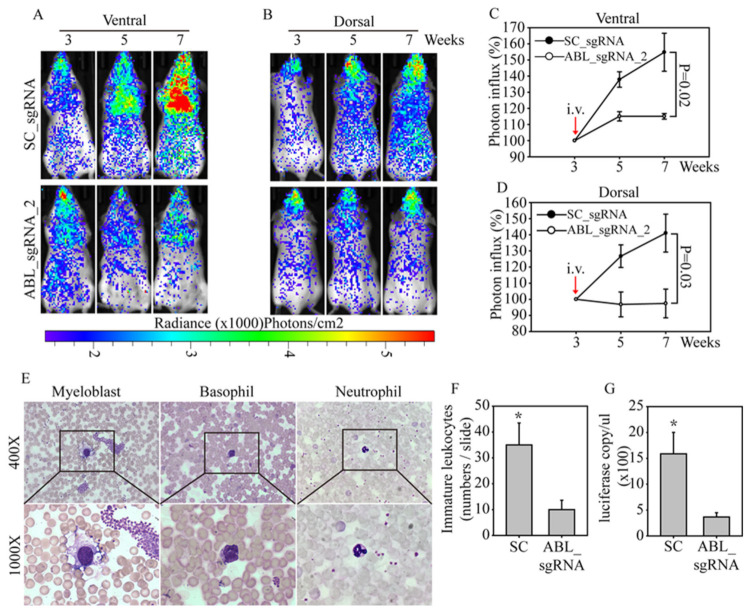
*ABL*-targeted CRISPR/Cas9 lentivirus therapy effectively inhibits leukemia cell growth in an animal model. For lentivirus therapy, 5×10^6^ luciferase-labeled K562 cells were injected via the tail vein into severe combined immunodeficient (SCID) mice. The mice were grouped (three mice/group) after three weeks and injected via the tail vein with SC- or *ABL*-targeting virus at a 100:1 ratio of virus (defined as MOI = 1) to the expected number of K562 cells (5 × 10^8^). Bioluminescence images were taken every two weeks with the mice in the (**A**) ventral and (**B**) dorsal positions. Bioluminescence images were analyzed by photon influx, which represents the number of human leukemia cells in the mice. The photon influx of each group of mice is presented and compared in both the (**C**) ventral and (**D**) dorsal positions. (**E**) Peripheral blood was collected from each mouse, and Liu’s stain was used to perform a WBC differential count. Myeloblasts, basophils and neutrophils are shown at 400× and 1000× magnification. (**F**) The immature WBC cell counts from the mice injected with SC- or ABL-targeting virus were analyzed. (**G**) DNA was purified from both groups of mice, and the luciferase gene expression was measured. All data were normalized to mouse *GUS* expression, which was used as a DNA input control. Data are presented as the mean and standard deviation. Data were analyzed with Student’s *t*-test; all *p*-values were two-sided. *p* values less than 0.05 are indicated with an asterisk.

**Figure 5 cancers-12-01399-f005:**
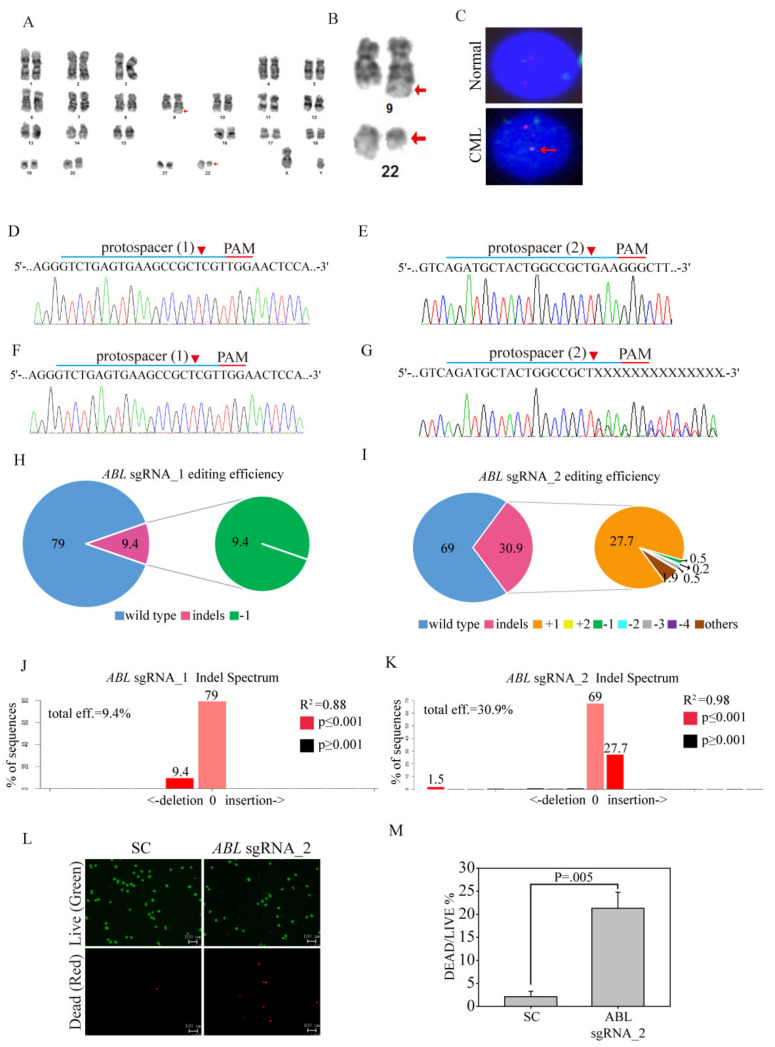
*Ex vivo ABL*-targeted CRISPR/Cas9 lentivirus therapy of CML patients. (**A**) Karyogram from the products of conception showing the karyotype of 46, XY, t(9;22)(q34;q11.2). (**B**) Highlight of chromosomes 9 and 22 at the same level of resolution. (**C**) Interphase fluorescence in situ hybridization (FISH) analysis using probes for the *BCR* (green) and *ABL* genes (red) shows an abnormal pattern t(9;22) of the fusion protein (yellow, lower panel) in the CML patient cells compared to the normal cells (separate colors). SC sgRNA and *ABL* sgRNAs were delivered to the clinical CML cells by lentivirus. After transduction, DNA from the virus-infected cells was purified and subjected to Sanger sequencing for the target sites of (**D**) *ABL* sgRNA_1 and (**E**) *ABL* sgRNA_2 in the SC sgRNA-transfected CML cells. (**F**) *ABL* sgRNA_1 and (**G**) *ABL* sgRNA_2 produced a mixture of sequences around the expected Cas9 cleavage site in a pool of gene-edited cells after *ABL* sgRNA lentivirus transduction. TIDE algorithm analysis of the *ABL* gene-edited sequence showed a high editing efficiency in clinical CML cells. The pie charts show the percentages of indels in the *ABL* gene edited by (**H**) *ABL* sgRNA_1 and (**I**) *ABL* sgRNA_2. The gene editing efficiency of the two sgRNAs is presented in red, while the two most common -1 and +1 indels are presented in green and orange, respectively. The original TIDE algorithm analysis is shown for (**J**) *ABL* sgRNA_1- and (**K**) *ABL* sgRNA_2 virus-transfected CML cells compared to SC-transfected cells. (**L**) The LIVE/DEAD cell viability assay was performed after SC sgRNA and *ABL* sgRNA_2 were delivered to the clinical CML cells by lentivirus. Cells were subjected to viability assays to identify live (green) and dead (red) cells. (**M**) Cell death after SC sgRNA and *ABL* sgRNA_2 lentivirus delivery was analyzed for significance. Data were analyzed with Student’s *t*-test; all *p*-values were two-sided.

**Figure 6 cancers-12-01399-f006:**
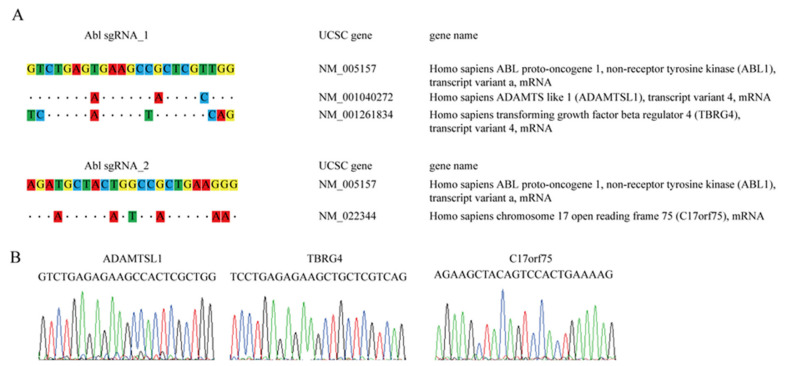
Off-target investigation of the *ABL-*targeted CRISPR/Cas9 system. (**A**) The CRISPR design website was used to predict off-target candidate genes for both the *ABL* sgRNA_1 and *ABL* sgRNA_2 viruses. Similarities are presented as dots, and mismatch sites are indicated by nucleotide substitution. (**B**) Sanger sequencing of the K562 cells infected with the *ABL* sgRNA_1 or *ABL* sgRNA_2 virus was used to examine potential indels in off-target candidate genes.

**Figure 7 cancers-12-01399-f007:**
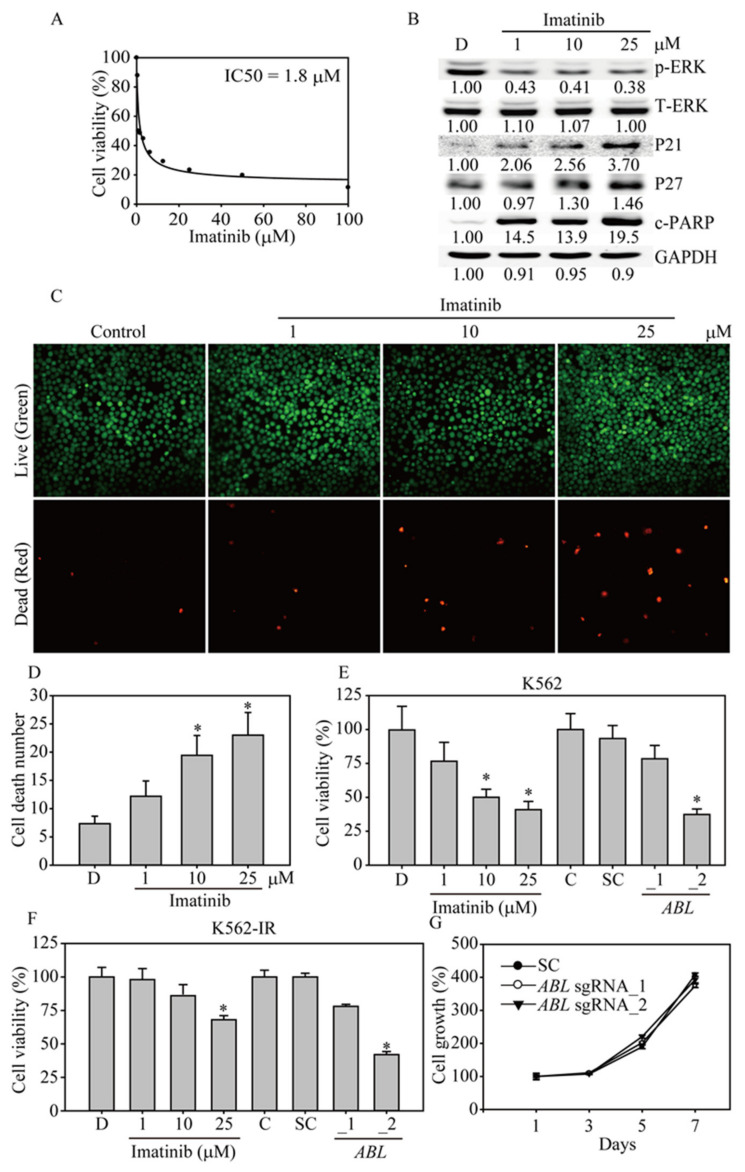
Imatinib inhibits K562 cell survival and induces apoptosis. (**A**) The IC50 values of the control or imatinib in K562 cells were determined using MTT assays after treatment for 48 h. (**B**) Imatinib significantly inhibited ERK activation and induced P21, P27 and cleaved PARP (c-PARP) protein expression in a dose-dependent manner, as evidenced by western blot analysis. All the western blotting was measured and quantified by Image J software. (**C**) The LIVE/DEAD cell viability assay was performed after imatinib treatment of K562 cells for 24 h. Cells were subjected to viability assays to identify live (green) and dead (red) cells at 100× total magnification (**D**) Cell death after imatinib treatment was analyzed for significance. Comparison of cell viability following imatinib treatment and *ABL* sgRNA virus infection of (**E**) K562 cells or (**F**) K562-IR cells. K562 or K562-IR cells were treated with 1, 10 or 25 μM imatinib or infected with *ABL* sgRNA_1 and *ABL* sgRNA_2 virus for 48 h. The cells were analyzed by MTT assays. (G) Cell viability curve of the *ABL* sgRNA_1- and *ABL* sgRNA_2-transduced HS27A cells determined by MTT assays. Data are presented as the mean and standard deviation. Data were analyzed with Student’s *t*-test; all P-values were two-sided. P values less than 0.05 are indicated with an asterisk.

**Figure 8 cancers-12-01399-f008:**
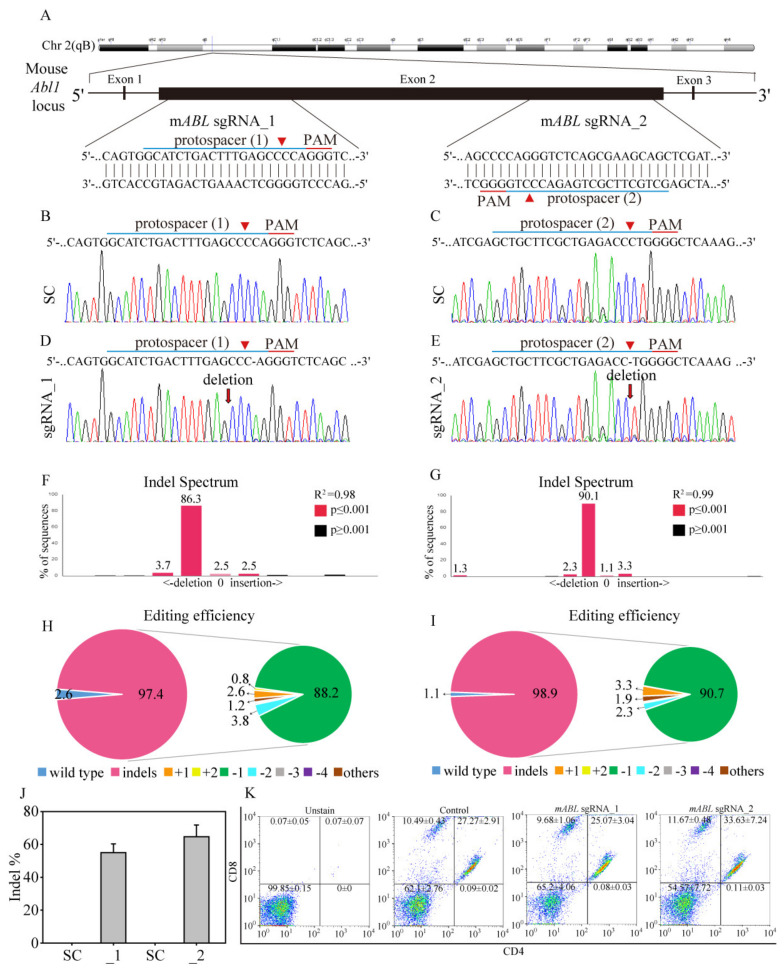
Mouse *ABL* gene targeting through CRISPR/Cas9 to investigate T-cell survival and development in vivo. (**A**) Schematic representation of the mouse *ABL* DNA locus and two protospacer sequences (blue underline) for editing. The arrowhead indicates the expected Cas9 cleavage site. Scrambled (SC) and mouse *ABL* (*mABL*) sgRNAs were delivered to NIH-3T3 cells by lentivirus. (**B**,**C**) After transduction, DNA from the virus-infected cells was purified and subjected to Sanger sequencing of the wild-type *mABL* DNA locus in NIH-3T3 cells. (**D**) *mABL* sgRNA_1 and (**E**) *mABL* sgRNA_2 produced a single nucleotide deletion around the expected Cas9 cleavage site in a pool of gene-edited cells after lentivirus transduction. TIDE algorithm analysis of the *mABL* gene-edited sequence (indels, insertions and deletions) showed a high editing efficiency in NIH-3T3 cells. The bar figures show the indel distribution of the (**F**) *mABL* sgRNA_1 and (**G**) *mABL* sgRNA_2 virus-transfected NIH-3T3 cells, compared to the SC transfected cells. The pie charts demonstrate the percentages of indels in the *mABL* gene edited by (**H**) *mABL* sgRNA_1 and (**I**) *mABL* sgRNA_2. The gene editing efficiency of the two sgRNAs is presented in red, while the most common -1 indel is presented in green color. (**J**) Gene edit efficiency (indel%) of the leukocytes was compared by SC-(*n* = 4), *mABL* sgRNA_1 (*n* = 3) and *mABL* sgRNA_2 (*n* = 3) introduced mice. (**K**) The representative T-cell subpopulations from three groups of mice were measured for leukocyte CD4 (FITC) and CD8 (PE) expressions by FACS counting. Data are presented as the mean and standard error.

**Figure 9 cancers-12-01399-f009:**
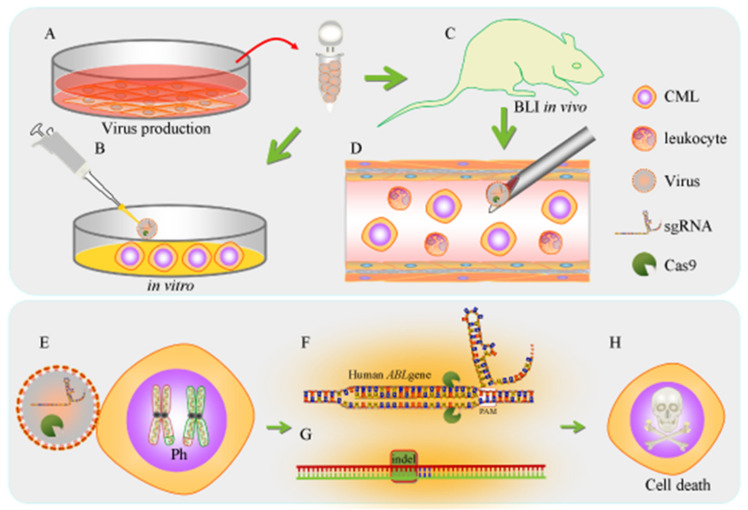
Schematic representation of the research strategy in this study. (**A**) Phoenix cells were used as the host to generate the lentiCRISPR plasmid-based *ABL* gene-edited virus. The collected lentivirus was purified, concentrated and quantitated by qPCR analysis. The high-quality lentivirus was used (**B**) in vitro and (**C**) in in vivo bioluminescence imaging-based animal models. (**D**) We injected lentivirus targeting bioluminescent human CML cells via the tail vein into the systemic leukemia xenograft mouse model. The leukemia cells were detected by IVIS. (**E**) Once the *ABL* gene knockout virus attacks the Ph chromosome in K562 cells, (**F**) the virus will generate sgRNA targeting the human *ABL* gene locus, and Cas9 will cleave the DNA. (**G**) The competing NHEJ pathway for DNA repair frequently creates indels. (**H**) CML cells eventually die due to the production of nonsense ABL sequences or early termination of the protein due to a frameshift.

**Table 1 cancers-12-01399-t001:** Cardiac puncture collected blood samples were taken from SC, *mABL* sgRNA_1 and *mABL* sgRNA_2 virus tail-vein injected mice and blood count parameters were quantified. No significant was found in these parameters.

Parameters	SC (*n* = 4)	*mABL* sgRNA_1 (*n* = 3)	*mABL* sgRNA_2 *n* = 3
RBC (×10^6^/uL)	9.72 ± 0.49	10.04 ± 0.3	10.04 ± 0.29
WBC (×10^3^/ uL)	3.77 ± 1.12	6.42 ± 1.06	4.24 ± 0.5
Hemoglobin (g/dL)	14.63 ± 0.79	15.17 ± 0.38	15.1 ± 0.42
Hematocrit (%)	45.83 ± 2.72	48.6 ± 1.4	47.13 ± 1.1
Neutrophils (%)	14.43 ± 1.22	13.07 ± 1.47	22.97 ± 4.8
Lymphocytes (%)	72.33 ± 4.76	82.4 ± 1.8	70.37 ± 6.96
Monocytes (U/Ul)	11.45 ± 5.56	10.1 ± 1.01	9.47 ± 1.91
Platelets (x10^3^/uL)	563.25 ± 130.27	521.67 ± 90.24	599.33 ± 129.12

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
