# Peer review of "ABL Genomic Editing Sufficiently Abolishes Oncogenesis of Human Chronic Myeloid Leukemia Cells In Vitro and In Vivo"

_cancers, 2020, doi:10.3390/cancers12061399_

Round 1

Reviewer 1 Report

The authors answered all the concerns and the manuscript is much better than the first version.

Author Response

Thanks for the review.

Reviewer 2 Report

The new version is fine for me.

Author Response

Thanks for the review.

This manuscript is a resubmission of an earlier submission. The following is a list of the peer review reports and author responses from that submission.

Round 1

Reviewer 1 Report

This article proposed method by targeting ABL gene through CRISPR/Cas9 instead of target BCR-ABL junction region to repress CML development. The authors verified their method in CML cell line, mouse model and patient sample and the results supported ABL gene editing-based therapy might provide a potential strategy for imatinib-insensitive or resistant CML patients. However, several concerns need to be answered.

Compare to target ABL gene, target BCR-ABL junction region may more efficient and feasible, because ABL gene has its own physiological roles such as regulating T‐cell development and mature T‐cell function, which is import for anti-cancer immune response. The variety of BCR-ABL junctions in CML patients is a challenge, but for a particular patient, the junction region is specific and easy to be defined nowadays, design specific sgRNA is not difficult. Moreover, there were at lest 2 papers reported to target CR-ABL junction region to repress CML cells by using CRISPR/Cas9, that impaired the novelty of this article. The authors directly injected concentrated virus to mice with very high MOI, what was the titer of the virus and what's final volume of virus suspension that was injected into mice? In Fig1I, why wild type ABL p145 was also downregulated even though the sgRNA was specific to BCR-ABL junction? For in vivo experiment, K562 cell line xenograft model is not enough and the mice number is too low. When the Authors try to demonstrate the advantage of their ABL editing to imatinib, imatinib resistant cell line is not a convincing control, any other treatment will show better effect than imatinib in imatinib resistant cell line, the authors can use CML LSC or mouse transplant model and analysis the residual cells to verify advantage of the ABL editing strategy.

Reviewer 2 Report

This is a nice paper about genomic editing of BCR-ABL in CML. I would suggest to modify the Figure 3 to show the apoptosis induction (by quantification of activated caspases for example) since data show only lack of proliferation and not apoptosis induction, as well described in the main text.

I am confused by WB images shown in Figures 3B and 1I since c-PARP looks like affected only in fig. 1I. If quantification could be reported it would be helpful for readers.

Do the authors have any additional explanation about the lack of effect on cell viability of the sg_1? The downstream signaling looks like be preserved but the proliferation rate is affected. We suggest, if possible, to test cell cycle phases distribution and eventually cyclin involvement in the phenotype.